# Pediatric snakebite in Sub-Saharan Africa: Clinical predictors, outcomes, and gaps in care—A systematic review

Samuel Mayeden[1], Margit Wirth[2], Nisreen Agbaria[1], Masood Ali Shaikh[3], Peter Dambach[1], Michael Lowery Wilson[4], Olaf Horstick[5], Salvador Camacho[2], Hans-Jörg Lang[1,5], Andreas Deckert[1]*

1 Heidelberg Institute of Global Health, Medical Faculty, Heidelberg University, Heidelberg, Germany, 2 Swiss Tropical and Public Health Institute, Basel, Switzerland, 3 Department of Medicine, College of Medicine, Korea University, Seoul, South Korea, 4 Heidelberg Institute of Global Health, Section of Oral Health, Medical Faculty, Heidelberg University, Heidelberg, Germany, 5 Heidelberg Institute of Global Health, Global Child Health, Medical Faculty, Heidelberg University, Heidelberg, Germany,

* a.deckert@uni-heidelberg.de

## Abstract

### Background

Snakebite envenomation (SBE) remains a significant but neglected public health problem among children in sub-Saharan Africa (SSA), with clinical predictors of severity and regional disparities in outcomes still poorly characterized.

### Methods

We conducted a systematic review of peer-reviewed studies published from data-base inception to October 2024 on pediatric snakebite envenomation (SBE) in SSA, focusing on clinical and demographic predictors of severe outcomes. Data on study characteristics, SBE symptoms, risk factors, and outcomes were synthesized descriptively.

### Results

Eighteen studies from six SSA countries were included (totaling 2,687 pediatric patients). Children affected were predominantly under 12 years, with a male predominance. Severe outcomes including death, amputation, and permanent disability were closely associated with delayed hospital presentation (> 6–12 hours), certain traditional first aid practices, young age (<10 years old), severe local swelling, upper limb bites, and laboratory. Most identified SBE syndromes were cytotoxic or haematotoxic; neurotoxic cases were rarely reported. Antivenom availability and type varied by country and facility; adverse reactions were common (≈20–70%, definition-dependent; anaphylaxis up to ~57%). Mortality ranged from <5% in South Africa and Kenya to 14% in Gambia and 36% in Cameroon, correlating with both antivenom and

**Data availability statement:** Data and code underlying this study are deposited in the hei-DATA repository (Heidelberg University) under https://doi.org/10.11588/DATA/S4SOH4. The dataset contains study-level aggregate data only (no individual patient data). Data license: CC0 1.0; code license: MIT.

**Funding:** The author(s) received no specific funding for this work.

**Competing interests:** The authors have declared that no competing interests exist.

essential emergency care access. Research gaps included limited data from several high-burden regions, lack of standardized pediatric protocols, and few long-term outcomes.

## Conclusions

Pediatric SBE in SSA is characterized by substantial preventable morbidity and mortality, mainly due to delays in care and inconsistent access to essential emergency care and safe antivenom. Improved health system capacity, rapid referral, and standardized management are urgently needed.

### Author summary

Snakebite envenoming remains a major but under-recognized cause of illness and death among children in sub-Saharan Africa. Children are particularly vulnerable due to their smaller body size, increased exposure to outdoor environments, and limited access to timely and appropriate medical care. This review synthesized all published studies on pediatric snakebite in the region to describe the affected populations, clinical management, and outcomes. Most cases involved boys under 12 years of age, typically bitten by *Echis ocellatus* and *Bitis arietans*, which are associated with severe local tissue damage, swelling, and coagulopathy. Mortality ranged from 6% to 8%, while permanent disability most often limb amputation or contracture occurred in approximately 4% of children, particularly where medical care was delayed or antivenom was unavailable. Neurotoxic envenoming was rarely reported, likely reflecting diagnostic challenges, under-recognition, or pre-hospital mortality. These findings underscore the urgent need to strengthen early referral systems, ensure consistent access to safe and effective antivenoms, and expand community education to reduce preventable deaths and long-term disability from snakebite among African children.

## 1. Introduction

Snakebite envenomation (SBE) remains a globally under-recognized yet significant public health problem, particularly in tropical and subtropical regions such as sub-Saharan Africa (SSA) and South-East Asia (SEA) [1,2]. According to World Health Organization (WHO) estimates, up to 5.4 million people are bitten by snakes annually, with approximately 2.7 million envenomings and up to 138,000 deaths worldwide [1,3]. However, these figures are likely underestimated, as many victims across SSA, SEA, and Latin America never reach hospitals, and their clinical information is often not captured in national health information systems [4,5]. Landmark pediatric snakebite studies from Asia and Latin America have similarly identified high risk among children for severe envenoming complications, delays to care, and inequitable access to antivenom [6–8]. These global findings highlight that pediatric snakebite is not

confined to SSA, but represents a widespread pediatric vulnerability across snakebite-endemic regions. Underreporting is further compounded by the inconsistent definition and recording of distinct SBE syndromes neurotoxic, cytotoxic, and haematotoxic presentations in clinical records [9,10].

Recognizing the scope of the problem, the WHO designated SBE as a Neglected Tropical Disease in 2017 and, in 2019, launched a comprehensive strategy to halve SBE deaths and disability by 2030 [1]. Despite these advances, there remains limited published evidence on the pediatric and maternal burden of SBE in SSA. Children and pregnant women are especially vulnerable due to physiological factors, increased risk exposures, and delays in receiving definitive care, defined as timely access to antivenom administration and supportive critical care within a health facility equipped to manage enveming-related complications [11,12]. Nevertheless, comparative evidence for delays in treatment of pediatric cases compared to adults is limited, and more research is needed [12].

Adequate SBE care, including timely access to essential critical and supportive interventions and regionally adapted antivenoms remains unavailable for many, due to health system limitations, geographic barriers, and socioeconomic constraints [5,13] The development, production, and distribution of antivenoms adapted to regional snake species remain major challenges in both Africa and Asia [14]. Strengthening care for SBE in emergency departments and critical care settings at all levels of training is crucial to improve outcomes [10]. Comparing patterns across global regions provides important context for understanding fundamental clinical risks in pediatric snakebite and underscores the need for regionally grounded treatment strategies [15,16].

Despite the commitment of a dedicated community of clinicians and researchers, significant evidence gaps in the identification of risk factors and patient outcomes persist, especially in the pediatric population [17,18]. Most published literature focuses on broad epidemiology or clinical manifestations, and only a few studies systematically examine which clinical, demographic, or health system-related factors are associated with poor outcomes among children [12].

This systematic review aims to address these gaps by synthesizing current evidence on key factors such as delayed presentation, age, clinical symptoms, traditional first aid practices and limited access to antivenom which are associated with severe clinical outcomes in pediatric SBE victims in SSA. Specifically, our objectives are: (1) to systematically identify and summarize clinical, demographic, and health system predictors of severe outcomes (including death, amputation, and permanent disability) among children with SBE; and (2) to map and evaluate the spectrum of reported outcomes to inform improvements in frontline clinical care, but also broader health system strengthening, resource allocation, and evidence-based policy development.

## 2.0 Methods

### 2.1 Search strategy

Our systematic review adhered to the Preferred Reporting Items for Systematic Reviews and Meta-Analyses (PRISMA) guidelines [19]. Two independent reviewers (MW, SM) conducted a comprehensive literature search across five databases: PubMed, SafetyLit, African Journals Online, CINAHL, and Google Scholar. The search aimed to identify all relevant studies published from database inception to October 2024 that reported pediatric snakebite envenomation in SSA. These databases were selected for their complementary coverage of international peer-reviewed biomedical research and region-specific African publications, ensuring a comprehensive capture of both globally indexed and locally sourced snakebite evidence. The initial search was performed on 1 August 2024, and was updated regularly, with the final search completed on 24 October 2024.

The search strategy was developed using three core concepts "children," "snakebite envenomation," and "sub-Saharan Africa" combined using Boolean operators. For each concept, we included a wide range of synonyms and related terms, such as "snakebite," "snake envenoming," and "snake venom poisoning" to ensure a comprehensive coverage of the available evidence in literature. In PubMed, we used Medical Subject Headings (MeSH) in addition to free-text terms. Due to database interface limitations, searches in Google Scholar and SafetyLit were conducted using split queries. The full search strings for each database are available in S1 Table.

We defined "children" as individuals up to their 18th birthday (i.e., < 18 years), in accordance with the United Nations Convention on the Rights of the Child [20]. SSA was defined according to the African Union's classification, which includes all African countries except those in North Africa.

Search results were exported to EndNote 20 and duplicates were identified and removed prior to screening. Titles and abstracts were independently screened (SM, MW, NA) using predefined inclusion and exclusion criteria. No language restrictions were imposed during the search process. However, all studies that met the inclusion criteria and included in the final review were published in English. Non-English records were screened at the title and abstract level where an English abstract was available; studies published without an English abstract or full text were excluded due to feasibility constraints.Only studies published in peer-reviewed journals were included, while non-peer-reviewed materials such as conference proceedings, theses, dissertations, and preprints were excluded. Peer-review status was verified using journal indexing metadata and publisher information. Where abstracts were unavailable, titles were retained for full-text review. Discrepancies were resolved through discussion and, if necessary, adjudication by a third reviewer (AD).

## 2.2 Inclusion and exclusion criteria

The inclusion criteria included peer-reviewed studies, and studies that addressed snakebite in children (<18 years old) in the SSA region, and provide data that were either exclusively pediatric or reported in a way that allowed for the extraction of child-specific results. Additional criteria also included that each study identified at least one clinical or demographic variable associated with severity, such as mortality, permanent disability, surgical intervention, or significant systemic complications. In order to maximize the breadth of the evidence base, we applied no restrictions based on study design, language, or year of publication.

Conversely, we excluded any record that did not fulfill the inclusion criteria. Studies were not considered if they had not undergone peer review or lacked systematic methodology. Studies were excluded if they were: non-peer reviewed, editorials, case reports, narrative reviews, thesis, preprints, or conference abstracts that were too limited in detail to allow critical appraisal. We also excluded studies that focused on populations outside of SSA, including North Africa, or on circumstances not relevant to pediatric care, such as snakebite in pregnancy. Any research that did not report pediatric-specific data, or from which the number of child participants could not be determined, was omitted. Authors where contacted for missing data or in cases where full texts were not available.

## 2.3 Data extraction

Data extraction was performed by the first reviewer (MW) using a standardized Excel template. Extracted data included author(s), year, country, study design, age, sample size, clinical characteristics of pediatric cases, risk factors for severity, antivenom use (including adverse reactions), and main outcomes. A second reviewer (SM) verified the extracted data. Any disagreements that emanated were resolved by consultation with a third reviewer (AD). Key characteristics and main findings of all included pediatric snakebite studies are summarized in S2 Table.

## 2.4 Effect measures

Effect measures extracted included odds ratios (ORs), adjusted odds ratios (AORs), case fatality rates (CFR), and the proportion of pediatric patients among our study population. The primary synthesis focused on reported clinical or demographic predictors of severe outcomes, including death, permanent disability, surgical intervention, or prolonged hospitalization.

## 2.5 Quality assessment

To ensure the credibility and reliability of our systematic review, three reviewers (MW, SM, NA) independently appraised the methodological quality of each included study and resolved any differences by discussion and when required with input from the third reviewer (AD). The Joanna Briggs Institute (JBI) Critical Appraisal Checklist was applied for case series studies [21]. For cross-sectional and cohort studies, the Strengthening the Reporting of Observational studies in

Epidemiology Statement (STROBE) [22]. was used. We used STARD 2015 for diagnostic accuracy studies [23], and TRI-POD for prediction model development or validation [24].

Studies were scored using each tool's criteria and classified as high, moderate, or low quality according to tertiles of the maximum possible score. No studies were excluded solely due to quality, but these ratings were central to our interpretation of literature. Quality assessment scores for all included studies are presented in S3 Table.

### 2.6 Synthesis and presentation of results

Each included study was assigned a unique reference number to support clear cross-referencing. Findings were synthesized using structured summary tables, figures, and descriptive statistics (frequencies, proportions). Due to substantial heterogeneity in study designs, populations, and outcome definitions, meta-analysis was not feasible. Instead, we conducted a narrative synthesis, integrating results across diverse settings. Studies were categorized as either exclusively pediatric or mixed-age cohorts (with disaggregated pediatric data) which enabled us to focus our data synthesis on child-specific predictors and outcomes.

## 3.0 Results

### 3.1 Identification and selection of studies

A total of 1,606 records were identified through databases searches. After removing duplicates, 981 records were screened by title and abstract, with 107 articles were assessed for full text. 18 studies met the inclusion criteria and were synthesed. The study selection process is summarized in Fig 1 (PRISMA 2020 flow chart; [19]).

### 3.2 Geographical distribution of included studies

A total of 18 studies on pediatric snakebite in SSA were included, spanning six countries with majority originated from South Africa (n = 8), followed by Nigeria (n = 3) and Ethiopia (n = 2), with single studies from Kenya, Gambia, and Cameroon. As shown in Fig 2, research activity is heavily concentrated in South Africa, while large parts of the region remain largely underrepresented. This uneven distribution underscores persistent gaps in the evidence base for pediatric snakebite across SSA, with many high-burden areas lacking published data to inform clinical practice and policy.

### 3.3 Quality assessment results

The methodological quality of all included studies varied considerably. Of the 18 studies assessed, 7 (39%) were rated as high quality, 8 (44%) as moderate quality, and 3 (17%) as low quality, based on scoring criteria for each study design and assessment tool S3 Table. Most high-quality studies originated from South Africa (n = 5), while studies from Nigeria (n = 1), Ethiopia (n = 1), Gambia (n = 0), and Cameroon (n = 0) were more often moderate or low quality.

### 3.4 Demographics and Bite Circumstances

A total of 2,687 pediatric snakebite envenomation patients were reported across all included studies. The pooled median age of affected children was 9 years (IQR 6–12; range 1–17 years),with a consistent male predominance ranging from 57% to 81% (Fig 3). Incidents showed strong seasonality, peaking during the rainy or warm months December to March in southern Africa and April to October in West Africa periods that align with increased agricultural activity and outdoor exposure. Most envenomation occurred while children were walking, playing, or farming; however, a significant proportion, particularly in Nigeria and Ethiopia, took place at night.

### 3.5 Delays and pre-hospital interventions

Timely access to medical care remains a substantial challenge. Median time to hospital presentation ranged from 5 hours (IQR 3–10) in some South African settings to as long as 20 hours (IQR 11–31) with over half of pediatric patients

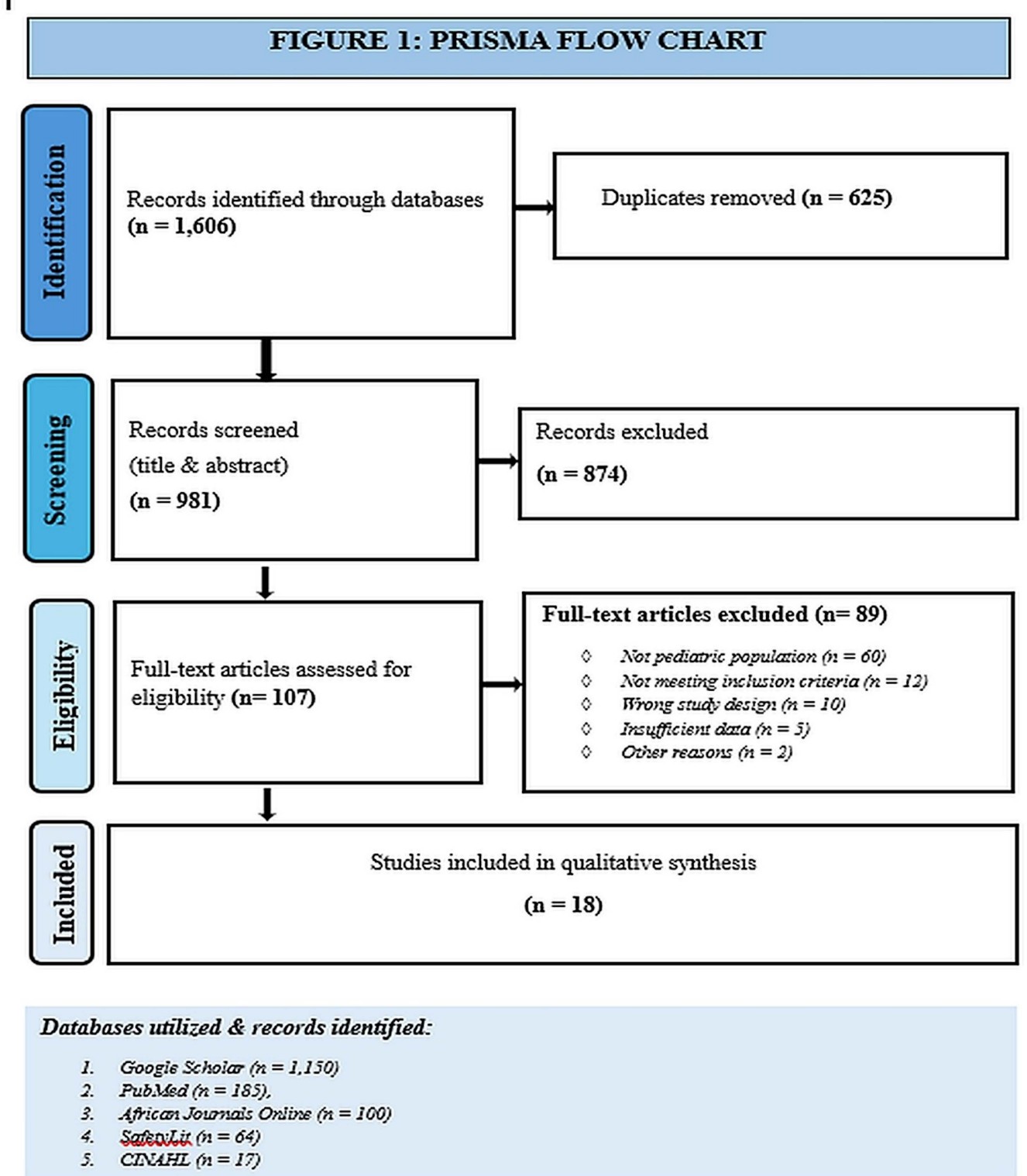

**Fig 1. PRISMA flow chart of study selection.** Flow diagram showing identification, screening, eligibility assessment, and inclusion of studies reporting pediatric snakebite in sub-Saharan Africa.

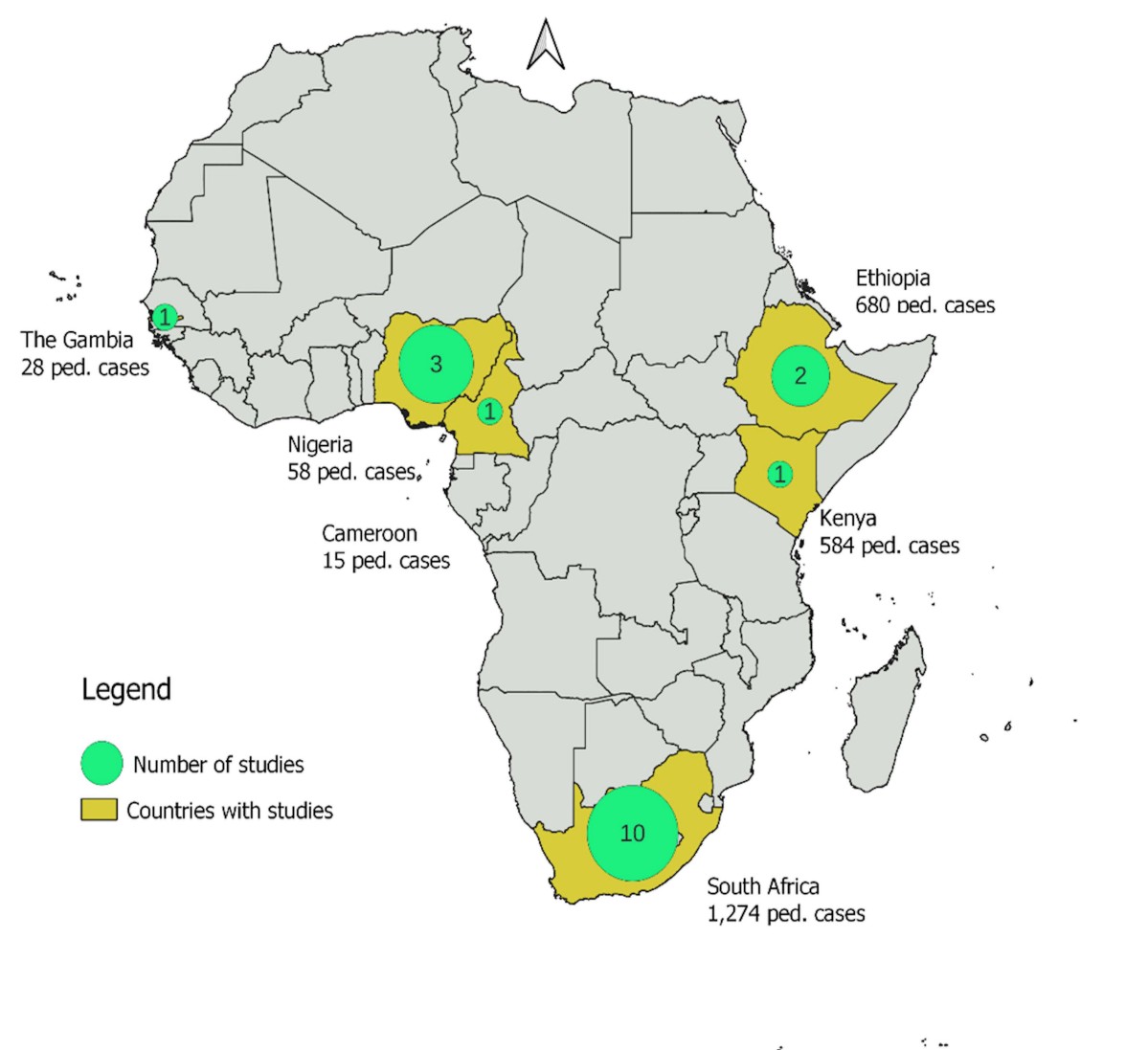

**Fig 2. Geographical distribution of pediatric snakebite studies in SSA.** Basemap source: Natural Earth (public domain data, https://www.natu-ralearthdata.com). Map data freely available under CC0 1.0 Public Domain license. Map showing the countries in sub-Saharan Africa where included pediatric snakebite studies were conducted.

experiencing delays greater than six hours (Fig 4). These delays were frequently exacerbated by harmful traditional first aid practices including tourniquets, incisions, herbal remedies, and traditional healer consultations which were especially prevalent in West and East Africa, documented in up to 80% of patients in Nigeria and Kenya. Such practices were strongly linked to both prolonged delays and the increased risk of complications. In Fig 4, an "X" indicates missing data for the parameter and country.

### 3.6 Clinical features and complications

Pain and swelling hallmarks of cytotoxic SBE were nearly universal, with lower limb involvement in up to 88% of cases (Fig 5). Severe local complications such as blistering, tissue necrosis, and suspected compartment syndrome were reported in up to 21% of patients, most frequently after delays in care or inappropriate first aid.

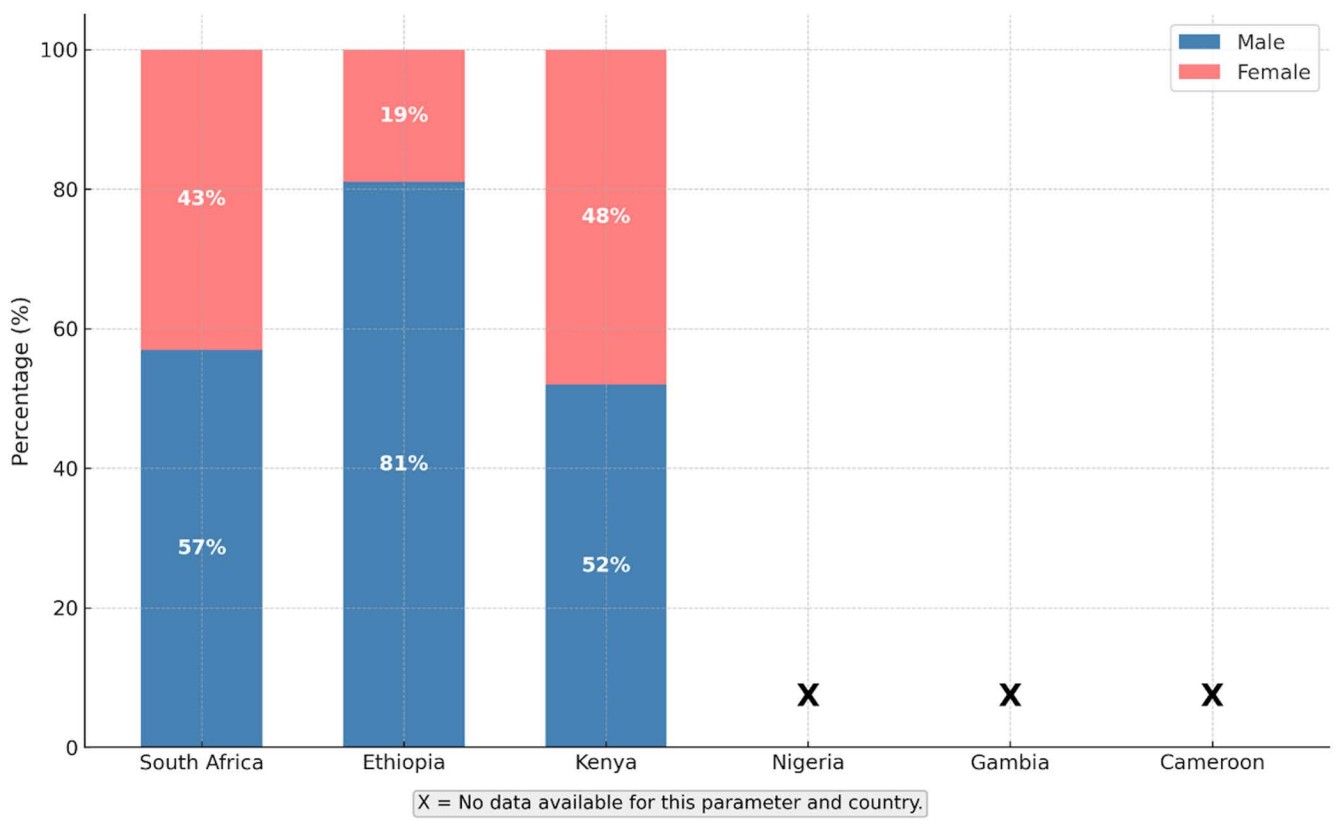

**Fig 3. Gender distribution of pediatric snakebite patients by country.** Proportions of male and female pediatric snakebite patients reported across included studies by country.

Systemic complications included shock, bleeding, coagulopathy, and acute kidney injury (AKI). These can result directly from the primary SBE, most notably with haematotoxic syndromes, which may overlap with cytotoxic features but may also arise secondarily due to critical hemorrhage, infected wounds, or pre-renal factors such as shock and delayed resuscitation. AKI was not always associated with the extent of local tissue damage, but rather with overall physiological risk and delays in receiving care.

A striking finding was also the *absence* of reported neurotoxic syndromes among the included studies. This warrants careful re-examination of the primary data, as such a pattern could reflect local snake species epidemiology, the impact of delayed access to care, or even unreported community-based mortality due to rapid progression of neurotoxic effects before hospital presentation. If confirmed, this epidemiological gap is significant and underscores the need for further surveillance and research into neurotoxic envenoming in SSA.

### 3.7 Antivenom use by country

The proportion of pediatric snakebite patients who received antivenom varied widely by country (Fig 6). In Nigeria, up to 92% of children received antivenom, compared to 11%–46% in South Africa, 25% in Kenya, and 21% in Gambia. In Ethiopia and Cameroon, antivenom was generally unavailable in S4 Table.

**Fig 4. Delayed hospital presentation and traditional first aid practices among pediatric snakebite patients by country.** Proportion of pediatric snakebite patients presenting late to hospital (>6 hours) and proportion reporting use of traditional first aid practices, by country.

### 3.8 Clinical Outcomes and prognostic indicators

Clinical outcomes varied considerably across regions, with mortality rates closely tied not only to antivenom availability, but also to the adequacy of supportive care and the timeliness of presentation (Figs 7A and 7B). The lowest mortality rates were observed in Kenya (1%) and South Africa (up to 4%), where children had relatively better access to both antivenom and essential supportive interventions. In Ethiopia and Nigeria, mortality ranged from 5% to 8%, while in Nigeria mortality range from 3% to 7.7%. Mortality was substantially higher in settings with limited or no antivenom availability, reaching 14% in Gambia and 36% in Cameroon.

Across included studies, approximately 60% of pediatric snakebite cases were attributed to identified snake species, while around 40% were reported without definitive species identification. A detailed summary of mortality causes and long-term sequelae reported across pediatric cohort is shown in S5 Table.

Amputation rates ranged from 0.5% to 6% across included cohort permanent disability was report as rare in South Africa and Kenya, while it was not consistently reported in other countries. Where available, median hospital stay ranged from 4 to 7 days. Additional outcome indicators and prognostic factors by country are summarized in S5 Table.

### 3.9 Predictors of severe outcomes

Fig 8 describe key predictors of severe outcomes in pediatric snakebite, displaying odds ratios and 95% confidence intervals for each clinical and laboratory risk factor. Delayed presentation (>6 hours), severe swelling or blistering, elevated INR, rural residence, traditional first aid, young age (<10 years), leukocytosis, and low hemoglobin were all associated with increased risk as shown in S6 Table. *(Note: Colors indicate variable categories; this is not a meta-analytic forest plot).*

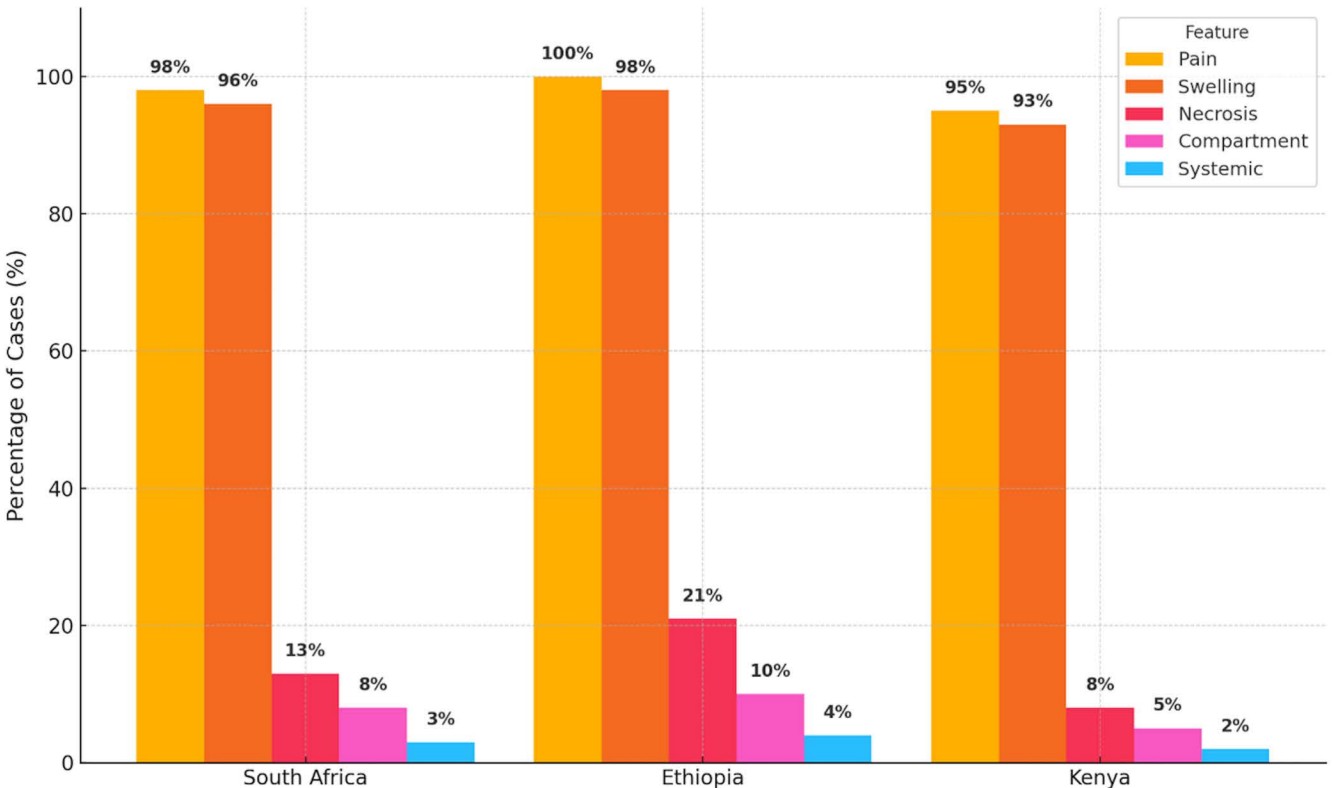

**Fig 5. Frequency of clinical features and complications among pediatric snakebite patients in SSA, by country.** Country-level distribution of reported clinical features and complications among pediatric snakebite patients across included studies.

## 4. Discussion

Our systematic review consolidates the available body of evidence on pediatric snakebite SSA, highlighting significant regional disparities, health system barriers, and clinical predictors of outcome. The findings confirm that snakebite remains a major, yet often neglected, cause of morbidity and mortality among children in SSA, consistent with global evidence [12,14]

Marked geographical differences were evident in both the scope and quality of the evidence base. The largest and most comprehensive studies originated from South Africa, where robust surveillance systems and greater access to both antivenom and surgical care likely contribute to improved outcomes [27–29,45]. In contrast, studies from Nigeria, Ethiopia, Gambia, Cameroon, and Kenya revealed substantial barriers including erratic antivenom supply, delayed presentation, and reliance on traditional healers which mirrored global patterns where antivenom access and health system readiness remain central determinants of outcome [3,5].

Regional variation in dominant snake species also influences clinical manifestations and antivenom efficacy. Cytotoxic and haematotoxic syndromes, largely due to viperid species (Bitis and Echis), predominate across SSA, though rear-fanged colubrids may also cause haematotoxicity. Elapid snakes (Naja, Dendroaspis, Hemachatus) are responsible for neurotoxic syndromes, often with overlapping cytotoxicity in spitting cobras [25]. Differences in regional snake ecology must be considered in syndrome-specific clinical training and antivenom selection.

Although no confirmed neurotoxic syndromes were reported among the pediatric cases in the included studies, this likely reflects diagnostic and reporting limitations, ecological variation in neurotoxic species distribution, and possible

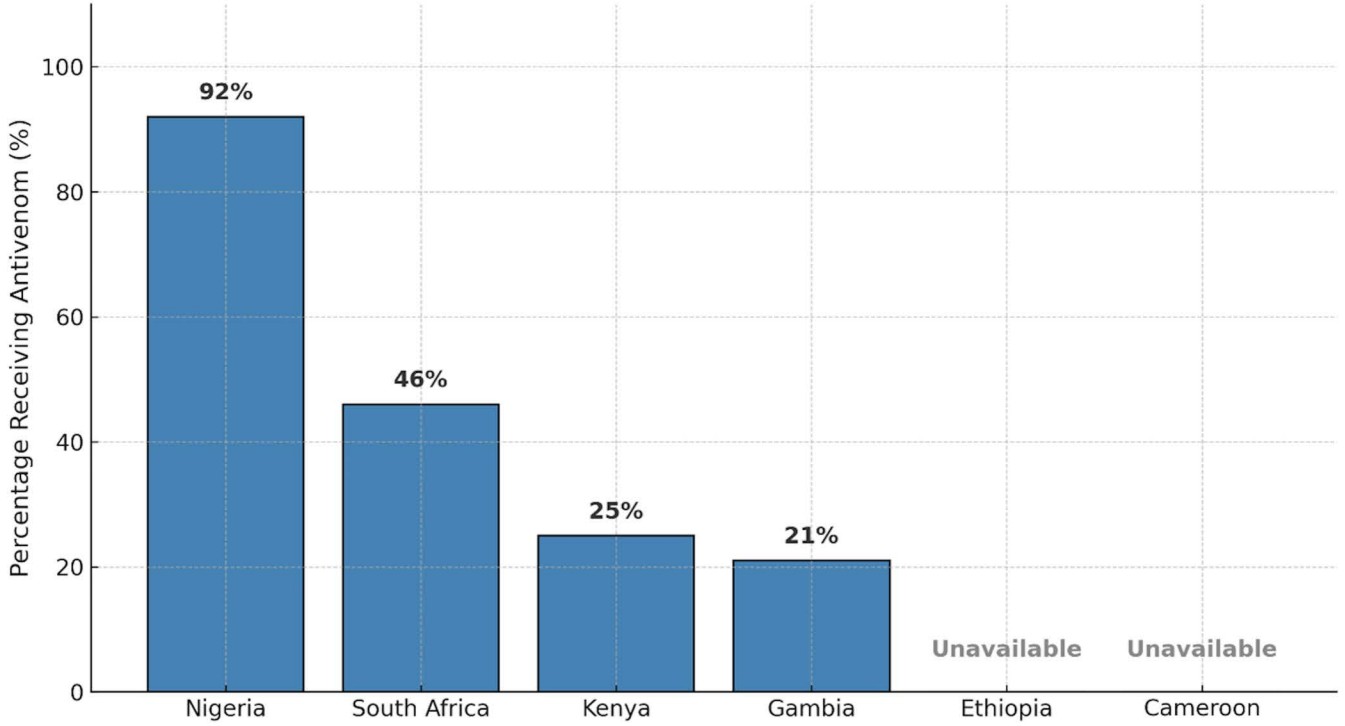

**Fig 6. Antivenom administration among pediatric snakebite patients by country.** Country-level proportions of pediatric snakebite patients who received antivenom, as reported in included studies.

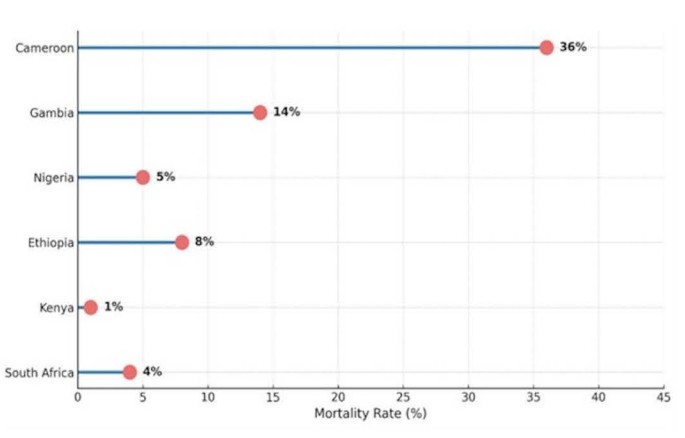

(A) Mortality rates among pediatric snakebite cohorts by country.

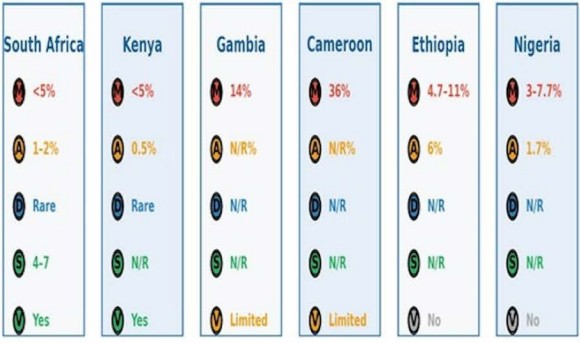

(B) Clinical outcome indicators by country

*Abbreviations: M, mortality; A, amputation; D, permanent disability; S, median hospital stays; V, antivenom access; N/R, not reported.*

**Fig 7. Country-level pediatric snakebite epidemiology and outcomes in SSA. (A)** Mortality rates among pediatric snakebite cohorts by country. **(B)** Clinical outcome indicators by country, including mortality, amputation, permanent disability, median hospital stay, and antivenom access. Abbreviations: M, mortality; A, amputation; D, permanent disability; S, median hospital stay; V, antivenom access; NR, not reported.

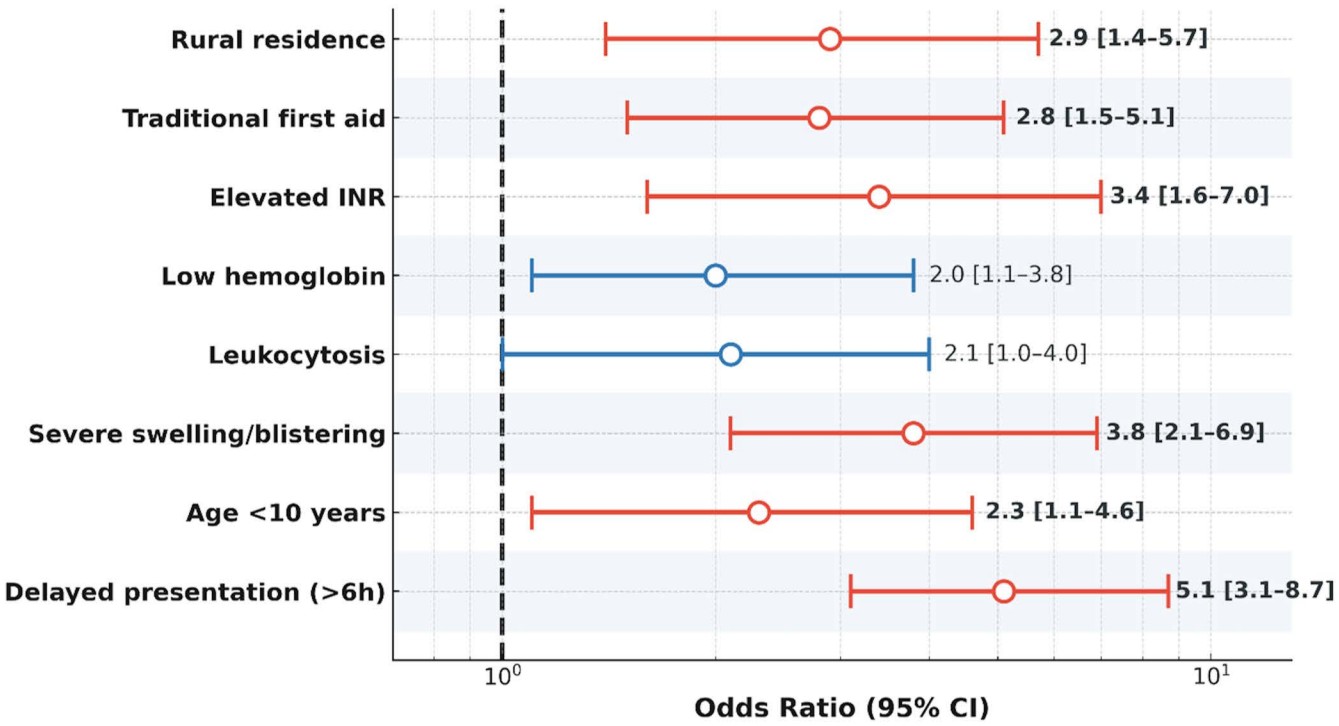

**Fig 8. Predictors of severe outcomes in pediatric snakebite.** Forest plot showing odds ratios and 95% confidence intervals for clinical and laboratory predictors of severe outcomes in pediatric snakebite. Colors indicate variable categories; this is not a meta-analytic forest plot.

pre-hospital mortality due to rapidly progressive respiratory failure [6,7,17], rather than a true absence of neurotoxic envenoming in SSA [8,26]. Neurotoxic pediatric snakebite has been documented elsewhere in the region, including envenoming by Naja annulifera and Dendroaspis polylepis in southern Africa.

Improved clinical recognition and routine documentation of neurological features in snakebite registries are therefore needed to define the true burden of neurotoxic syndromes and to support appropriate antivenom planning [25,30,31].

Identifying the predominant snake species responsible for pediatric envenoming in specific regions also has important public health implications. Knowledge of whether cytotoxic viperid species or neurotoxic elapids predominate in a given area can guide the development of community prevention strategies, risk-communication messaging, and awareness campaigns tailored to local ecology. Region-specific health education including recognition of high-risk species and behaviors associated with exposure may reduce snakebite incidence among children and improve early care-seeking [26].

To provide an international context, our findings in SSA can be compared with trends observed in other regions where pediatric snakebite burden is high. In South and Southeast Asia, for example, mortality among children treated in hospital settings typically ranges from 2–5% [15,31], substantially lower than the average mortality observed in SSA (6–8%), likely reflecting earlier presentation and broader antivenom availability [15,16,26]. In Latin America, despite different snake fauna dominated by *Bothrops* and *Crotalus* species, necrosis-related disability patterns among survivors mirror those seen in SSA, with amputation, limb deformity, and contractures representing major contributors to long-term morbidity [32,33]. These cross-regional comparisons underscore the importance of contextualizing SSA outcomes within global experiences and suggest that improvements in pre-hospital care, antivenom access, and early clinical recognition as implemented in parts of Asia may offer feasible models to reduce mortality in SSA pediatric populations [6,34].

Delays in hospital presentation, often exceeding 6–12 hours, were widespread, especially in Nigeria, Ethiopia, and Kenya [35–39]. Harmful traditional first aid practices frequently compounded these delays, with well-documented links to adverse outcomes [35,36,38–40]. These findings underscore the need for improved community education, strengthened referral systems, and rapid access to effective care across the levels of care.

Clinical features were heterogeneous, but severe local complications and systemic effects were more common following delayed presentation or inappropriate first aid [28,41]. Importantly, compartment syndrome diagnosis should be approached cautiously in this context, as reliance on clinical assessment alone may overestimate the need for fasciotomy, given that intra-compartmental pressures are often not markedly elevated [9,44]. Predictors of poor outcome included delayed care, younger age, severe swelling (reflecting cytotoxic SBE syndromes), laboratory evidence of coagulopathy or anemia (haematotoxic syndromes) [43], and upper limb bites. Notably, no neurotoxic SBE syndromes were reported among the included studies. This absence could reflect underlying snake ecology, unrecognized or unreported pre-hospital mortality, or gaps in clinical recognition. Re-examination of primary data and further prospective research are warranted, as these patterns have major implications for anti-venom selection, training, and health policy development [9,10].

The overall methodological quality of included studies was moderate to high, particularly among studies from South Africa, where design and reporting standards were more robust. However, several persistent limitations were identified including incomplete follow-up, inconsistent definitions, and underreporting of adverse events or long-term outcomes. Studies from other regions showed greater variability in quality. These methodological issues, together with the predominance of retrospective, single-center studies and substantial heterogeneity in design and outcome definitions, limit the certainty and generalizability of our conclusions.

While antivenom remains the only specific therapy for snakebite envenoming, its use varied widely ranging from high but targeted administration in South Africa and Kenya to minimal or absent use in Cameroon and Ethiopia [6,36,41,42]. Data on product types, dosing, titration, and infusion protocols were limited and inconsistently reported. Adverse reactions, including anaphylaxis, were not uncommon and may be more frequent in children, though this remains to be established [36]. Optimal outcomes rely not only on antivenom, but also on comprehensive supportive and surgical care including airway management, fluid resuscitation, monitoring, wound and infection management, and timely surgical intervention were indicated [9,10]

Several limitations were considered. Most studies were retrospective, single-center, and hospital-based, with potential selection bias and limited generalizability. Substantial heterogeneity in study design, outcome definitions, and reporting restricted direct comparison and precluded meta-analysis. Long-term outcomes, disability, and psychosocial effects were rarely reported. Gaps in surveillance, especially in under-resourced settings, led to underestimation of true incidence and severity. Although no language limits were imposed and several French-language studies were screened, only those with English abstracts or summaries were retrieved so studies published without any English summary may have been missed.

These findings reinforce the need for multi-center, prospective studies, standardized data collection, and improved health system capacity. Further research should clarify anti-venom dosing and administration protocols and health system strategies to reduce delays in care, in alignment with WHO-defined antivenom target product profiles for SSA [7,26].

A One Health approach which recognizes the interconnectedness of human, animal, and environmental health should be considered in the development of comprehensive prevention and intervention strategies [5]. Snakebite risk is shaped by ecological and human factors, including environmental context, community behaviors, socioeconomic status, and parental occupation. Locally adapted preventive measures such as improved housing, the use of light and mosquito nets, and community education may benefit from interdisciplinary and cross-sectoral collaboration [2,6]. While this systematic review focuses on SSA, cross-learning from experiences in other LMICs, including Southeast Asia and Latin America, will further support a holistic and sustainable solutions. Addressing these gaps will require coordinated efforts from clinicians, public health professionals, policy makers, and the broader community support systems [2,32]. It is worth highlighting that

de novo designed proteins capable of neutralizing diverse snake venom toxins in animal models represent a paradigm shift in SBE-specific therapeutics, offering potential solutions to many limitations of conventional antivenoms [30].

Consistent with earlier discussion, the absence of reported neurotoxic syndromes in the included studies is most plausibly explained by underrecognition, diagnostic limitations, ecological variation, or pre-hospital mortality due to rapidly progressive paralysis, rather than a true absence of neurotoxic envenoming in SSA [6,7].

## 5. Conclusions

Pediatric snakebite in SSA remains a substantial but under-recognized public health challenge, disproportionately affecting children in rural and resource-limited areas. Persistent barriers such as delayed hospital presentation, limited access to effective antivenom, and insufficient capacity for pediatric emergencies and critical care continue to drive high rates of morbidity and mortality. Outcomes are further shaped by gaps in health system infrastructure, the use of certain traditional first aid practices, and a lack of standardized protocols.

Addressing these challenges requires urgent, coordinated investment in health system strengthening, including consistent anti-venom supply, training in evidence-based management of all snakebite envenoming syndromes, and capacity for supportive and surgical care. There is a pressing need to improve community awareness, promote safe first aid, and build robust referral networks, particularly in rural settings. Standardizing clinical protocols and enhancing surveillance will enable more accurate monitoring of incidence and outcomes. Further research, especially multi-center and prospective studies, should focus on optimal antivenom dosing, management of SBE syndromes, and long-term outcomes for children. Finally, regional and international collaboration is essential to ensure all children at risk receive timely, equitable, and effective care.

## Supporting information

**S1 Table. Detailed search strategies for all databases.** Full electronic search strings used across all databases for identification of pediatric snakebite envenoming studies in sub-Saharan Africa.
(DOCX)

**S2 Table. Key characteristics and findings of all included articles.** Summary of extracted study characteristics, including country, study design, sample size, key clinical features, management, and main outcomes reported in included pediatric snakebite studies.
(DOCX)

**S3 Table. Quality assessment scores of all included articles.** Methodological quality appraisal results for included studies using standardized assessment criteria, categorized as high, moderate, or low quality.
(DOCX)

**S4 Table. Antivenom use, products, and references by country.** Country-level summary of antivenom availability and use, including reported products and supporting references from included studies.
(DOCX)

**S5 Table. Country-level pediatric snakebite epidemiology, clinical outcomes, and prognostic indicators in SSA.** Summary of country-level epidemiologic patterns and clinical outcomes, including mortality, amputation, permanent disability, median hospital stay, and key prognostic indicators reported across included studies.
(DOCX)

**S6 Table. Predictors of severe outcomes in pediatric snakebite in SSA.** Summary of clinical and laboratory predictors of severe outcomes, including odds ratios and 95% confidence intervals, as presented in the narrative synthesis.
(DOCX)

## Acknowledgments

The authors would like to thank all researchers and clinicians whose work contributed to the studies included in this systematic review. We extend our sincere gratitude to Dr. John Koku Awoonor, Technical Advisor, Ministry of Health, Ghana, and Prof. Alfred Yawson, Provost of the University of Ghana Medical School, for their invaluable support and insights. For the publication fee we acknowledge fnancial support by Heidelberg University

## Author contributions

**Conceptualization:** Samuel Mayeden, Margit Wirth, Andreas Deckert.

**Formal analysis:** Samuel Mayeden, Margit Wirth, Andreas Deckert.

**Methodology:** Samuel Mayeden, Margit Wirth, Nisreen Agbaria, Andreas Deckert.

**Project administration:** Margit Wirth, Andreas Deckert.

**Supervision:** Olaf Horstick, Salvador Camacho, Hans-Jörg Lang, Andreas Deckert.

**Validation:** Hans-Jörg Lang.

**Visualization:** Samuel Mayeden, Peter Dambach, Andreas Deckert.

**Writing – original draft:** Samuel Mayeden, Andreas Deckert.

**Writing – review & editing:** Samuel Mayeden, Nisreen Agbaria, Masood Ali Shaikh, Peter Dambach, Michael Lowery Wilson, Olaf Horstick, Salvador Camacho, Hans-Jörg Lang, Andreas Deckert.

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
