## [Decision Letter · Decision Letter 0]

1 Oct 2025

Pediatric Snakebite in Sub-Saharan Africa: Clinical Predictors, Outcomes, and Gaps in Care—A Systematic Review

Dear Dr. Mayeden,

Thank you for submitting your manuscript to PLOS Neglected Tropical Diseases. After careful consideration, we feel that it has merit but does not fully meet PLOS Neglected Tropical Diseases's publication criteria as it currently stands. Therefore, we invite you to submit a revised version of the manuscript that addresses the points raised during the review process.

Please submit your revised manuscript within 60 days Nov 30 2025 11:59PM. If you will need more time than this to complete your revisions, please reply to this message or contact the journal office at plosntds@plos.org. Please include the following items when submitting your revised manuscript:

We look forward to receiving your revised manuscript.

Kind regards,

Wuelton Monteiro, Ph.D.

Section Editor

Wuelton Monteiro

Section Editor

Shaden Kamhawi

co-Editor-in-Chief

Paul Brindley

co-Editor-in-Chief

**Journal Requirements:**

At this stage, the following Authors/Authors require contributions: Samuel Mayeden, Margit Wirth, Nisreen Agbaria, Masood Ali Shaikh, Peter Dambach, Michael Lowery Wilson, Olaf Horstick, Salvador Camacho, Hans-Jörg Lang, and Andreas Deckert. Please ensure that the full contributions of each author are acknowledged in the "Add/Edit/Remove Authors" section of our submission form.

Potential Copyright Issues:

- Figure 2. Please (a) provide a direct link to the base layer of the map (i.e., the country or region border shape) and ensure this is also included in the figure legend; and (b) provide a link to the terms of use / license information for the base layer image or shapefile. We cannot publish proprietary or copyrighted maps (e.g. Google Maps, Mapquest) and the terms of use for your map base layer must be compatible with our CC BY 4.0 license.

**Reviewers' Comments:**

Reviewer's Responses to Questions

**Key Review Criteria Required for Acceptance?**

**Methods**

-Are the objectives of the study clearly articulated with a clear testable hypothesis stated?

-Is the study design appropriate to address the stated objectives?

-Is the population clearly described and appropriate for the hypothesis being tested?

-Is the sample size sufficient to ensure adequate power to address the hypothesis being tested?

-Were correct statistical analysis used to support conclusions?

-Are there concerns about ethical or regulatory requirements being met?

Reviewer #1: Objectives clear

Appropriate design

Methods good - missing PROSPERO registration a priori

Sample size N/A

Correct stats and no ethical issues

Reviewer #2: NO

Reviewer #3: -N/A

-Yes

-N/A

-N/A

-N/A

-N/A

**Results**

-Does the analysis presented match the analysis plan?

-Are the results clearly and completely presented?

-Are the figures (Tables, Images) of sufficient quality for clarity?

Reviewer #1: Standard reporting - correct and figures well designed

Reviewer #2: NO

Reviewer #3: -Yes

-Yes

-Yes

**Conclusions**

-Are the conclusions supported by the data presented?

-Are the limitations of analysis clearly described?

-Do the authors discuss how these data can be helpful to advance our understanding of the topic under study?

-Is public health relevance addressed?

Reviewer #1: Appropriate

Reviewer #2: NO

Reviewer #3: -Yes

-N/A

-Yes

-Yes

**Editorial and Data Presentation Modifications?**

Reviewer #1: Change GENDER to SEX throughout as per SAGER guidelines of ICJME

Ref 31 is incorrect - first author is Wood D, not Darryl W.

Reviewer #2: NO

Reviewer #3: (No Response)

**Summary and General Comments**

Reviewer #1: This is a useful systematic review, however the context is not given in the Discussion. The differences between species across the region and therefore the need for antivenom are very different, for examples SA does not have Echis, which is common in central SSA, for which there is a specific antivenom, while most SA bites are not AV requiring. The lack of "neurotoxic" bites mentioned in the discussion is inaccurate as a number of the papers from Wood et al. include neurotoxic cases. See S Afr Med J 2009; 99: 814-818. Also see S Afr Med J 2017;107(12):1082-1085. DOI:10.7196/SAMJ.2017.v107i12.12628 which focusses on cytotoxic bites.

Reviewer #2: Dear editor,

This is an excellent systematic review with rigorous inclusion criteria, a substantial number of cases, and generally reliable literature quality. It provides a comprehensive analysis of the clinical characteristics of pediatric snakebite envenomation. However, I would suggest three points for potential improvement:

Primary Causative Snake Species: While we do not advocate using snake species as the primary criterion for treatment or management, but identifying the predominant snake species responsible for envenomation would hold significant public health value for local prevention and awareness campaigns.

Mortality and Disability Analysis: If feasible, I suggest supplementing the review with a more detailed discussion or synthesis of the causes of mortality and long-term disability, where available data permits

Age: The statement that “The affected children were predominantly under 12 years of age” is somewhat broad. Could you specify the exact age range, median, or average age of snakebite?

Thank you very much!

Rongde Lai(Emergency Department, the First Affiliated Hospital of Guangzhou Medical University)

2025-9-22

Reviewer #3: Pediatric Snakebite in Sub-Saharan Africa: Clinical Predictors, Outcomes, and Gaps in Care—A Systematic Review

This is an important systematic review because studies on paediatric snakebites are few in number. However, some deficiencies should be corrected, especially in methodology, as pointed out below.

Abstract

This should be rewritten concerning the Methods

Methods

….published from……..to …..

Need to include the duration

Age should be defined

what languages used?

Databases and keywords should be included.

Introduction

It is better to include contents of some landmark paed snakebite papers other than in SSA

…….definitive care defined as defined as timely…..

Methods

What is the basis of selecting only 5 databases?

What languages were used? This should be included in to the inclusion/exclusion criteria

Inclusion/exclusion criteria

How did the authors confirm that selected papers are peer-reviewed or not?

Figure 3, 4, 5 and 7

The legend should be placed below the figure.

Results

How many known snakebites and unknown snakebites?

In known snakebites, classify them according to the snake species

The systemic manifestations have been quantified as 3%, 4% and 2% in major SSA countries. What are those manifestations? Include their numbers. This should be discussed in the Discussion.

……A striking finding was also the absence of reported neurotoxic syndromes among the included studies.

In practical point of view, this can not happen because in this region, there are neurotoxic snakes. Please clarify this.

Discussion

It is better to compare SSA results with results of other part of the world. Other than that, the discussion is OK.

PLOS authors have the option to publish the peer review history of their article (what does this mean? ). If published, this will include your full peer review and any attached files.

**Do you want your identity to be public for this peer review?** For information about this choice, including consent withdrawal, please see our Privacy Policy .

Reviewer #1: No

Reviewer #2: No

Reviewer #3: No

**Figure resubmission:**
---

## [Decision Letter · Decision Letter 1]

6 Jan 2026

Pediatric Snakebite in Sub-Saharan Africa: Clinical Predictors, Outcomes, and Gaps in Care—A Systematic Review

Dear Dr. Mayeden,

Thank you for submitting your manuscript to PLOS Neglected Tropical Diseases. After careful consideration, we feel that it has merit but does not fully meet PLOS Neglected Tropical Diseases's publication criteria as it currently stands. Therefore, we invite you to submit a revised version of the manuscript that addresses the points raised during the review process.

Please submit your revised manuscript within by Feb 05 2026 11:59PM. If you will need more time than this to complete your revisions, please reply to this message or contact the journal office at plosntds@plos.org. Please include the following items when submitting your revised manuscript:

We look forward to receiving your revised manuscript.

Kind regards,

Wuelton Monteiro, Ph.D.

Section Editor

Wuelton Monteiro

Section Editor

Shaden Kamhawi

co-Editor-in-Chief

Paul Brindley

co-Editor-in-Chief

**Journal Requirements:**

1) Please upload all main figures as separate Figure files in .tif or .eps format. For more information about how to convert and format your figure files please see our guidelines:

2) We have noticed that you have uploaded Supporting Information files, but you have not included a list of legends. Please add a full list of legends for your Supporting Information files after the references list.

3) We notice that your supplementary figures are uploaded with the file type 'Table'. Please amend the file type to 'Supporting Information'. Please ensure that each Supporting Information file has a legend listed in the manuscript after the references list.

**Reviewers' Comments:**

Reviewer's Responses to Questions

**Key Review Criteria Required for Acceptance?**

**Methods**

-Are the objectives of the study clearly articulated with a clear testable hypothesis stated?

-Is the study design appropriate to address the stated objectives?

-Is the population clearly described and appropriate for the hypothesis being tested?

-Is the sample size sufficient to ensure adequate power to address the hypothesis being tested?

-Were correct statistical analysis used to support conclusions?

-Are there concerns about ethical or regulatory requirements being met?

Reviewer #1: All changes acceptable

Reviewer #2: -

Reviewer #3: (No Response)

**Results**

-Does the analysis presented match the analysis plan?

-Are the results clearly and completely presented?

-Are the figures (Tables, Images) of sufficient quality for clarity?

Reviewer #1: All changes acceptable

Reviewer #2: -

Reviewer #3: (No Response)

**Conclusions**

-Are the conclusions supported by the data presented?

-Are the limitations of analysis clearly described?

-Do the authors discuss how these data can be helpful to advance our understanding of the topic under study?

-Is public health relevance addressed?

Reviewer #1: All changes acceptable

Reviewer #2: -

Reviewer #3: (No Response)

**Editorial and Data Presentation Modifications?**

Reviewer #1: All changes acceptable

Reviewer #2: -

Reviewer #3: (No Response)

**Summary and General Comments**

Reviewer #1: All changes acceptable

Reviewer #2: -

Reviewer #3: Pediatric Snakebite in Sub-Saharan Africa: Clinical Predictors, Outcomes, and Gaps in Care-A Systematic Review

Most of my queries have been addressed; however, a few revisions are still required.

Line 172-176

If no language restrictions were applied, were there any articles published in languages other than English? If so, which languages were included, and how were their contents incorporated into the review?

Results

This section includes information that should be moved to the Discussion. Some examples are indicated below. Please correct these.

Line 276-278

Importantly, compartment syndrome diagnosis should be approached cautiously in this context, as clinical assessment may overestimate the need for fasciotomy and intra-compartmental pressures are often not markedly elevated [9]

Line 313-317

Among known snakebites, most were due to cytotoxic and haematotoxic viperid species (Echis, Bitis), consistent with regional epidemiology [25]. A smaller proportion of bites were attributed to neurotoxic elapids (Naja, Dendroaspis), although neurotoxic envenoming may be under-recognized in pediatric data due to diagnostic limitations and pre-hospital mortality [6,7,17].

Line 328-330

…..most commonly resulting from haemodynamic instability (septic or haemorrhagic shock) and respiratory failure, consistent with recognized mortality mechanisms in pediatric envenoming [25,26].

PLOS authors have the option to publish the peer review history of their article (what does this mean? ). If published, this will include your full peer review and any attached files.

**Do you want your identity to be public for this peer review?** For information about this choice, including consent withdrawal, please see our Privacy Policy .

Reviewer #1: No

Reviewer #2: No

Reviewer #3: No

**Figure resubmission:**
---

## [Editor Report · Decision Letter 2]

30 Jan 2026

Dear Mayeden,

We are pleased to inform you that your manuscript 'Pediatric Snakebite in Sub-Saharan Africa: Clinical Predictors, Outcomes, and Gaps in Care—A Systematic Review' has been provisionally accepted for publication in PLOS Neglected Tropical Diseases.

Best regards,

Wuelton Monteiro, Ph.D.

Section Editor

Wuelton Monteiro

Section Editor

Shaden Kamhawi

co-Editor-in-Chief

Paul Brindley

co-Editor-in-Chief

---

## [Editor Report · Acceptance letter]

Dear Dr. Deckert,

We are delighted to inform you that your manuscript, "

Pediatric Snakebite in Sub-Saharan Africa: Clinical Predictors, Outcomes, and Gaps in Care—A Systematic Review," has been formally accepted for publication in PLOS Neglected Tropical Diseases.

Best regards,

Shaden Kamhawi

co-Editor-in-Chief

Paul Brindley

co-Editor-in-Chief
